# Damper Winding for Noise and Vibration Reduction of a Permanent Magnet Synchronous Machine

**DOI:** 10.3390/s22072738

**Published:** 2022-04-02

**Authors:** Sijie Ni, Grégory Bauw, Raphaël Romary, Bertrand Cassoret, Jean Le Besnerais

**Affiliations:** 1Laboratoire Systèmes Electrotechniques et Environnement, University of Artois, UR 4025, F-62400 Béthune, France; gregory.bauw@univ-artois.fr (G.B.); raphael.romary@univ-artois.fr (R.R.); bertrand.cassoret@univ-artois.fr (B.C.); 2Eomys Engineering, F-59260 Lille, France; jean.lebesnerais@eomys.com

**Keywords:** PMSM, damper winding, noise, harmonic, passive, reduction, PWM

## Abstract

In this paper, a passive method for the noise reduction of the PMSM (Permanent Magnet Synchronous Machine) is presented. The principle is to add an auxiliary three-phase winding into the same slots as the initial stator winding, short-circuited via three capacitors of suitable values. The aim is to create a damping effect for flux density harmonic components, especially high-frequency harmonics from the PWM (PulseWidth Modulation), in the air gap in order to reduce the noise and vibration of the PMSM. The method can significantly reduce the global sound pressure level and vibrations for specific frequencies. Because of passive features, the additional winding effectively mitigates magnetic noise without greatly increasing the complexity of design and manufacturing, which also extends its applicability to different PMSMs.

## 1. Introduction

With the use and development of clean energy, electrification has been able to be rapidly deployed in various fields. PMSMs, on account of their high efficiency and high power density, are becoming increasingly important in the sectors of manufacturing and transportation. New problems have appeared, such as the electromagnetic noise and vibration in PMSMs [1,2,3,4]. The noise signature of PMSMs deserves further attention and research in order to improve the comfort of human hearing.

Usually, mechanical factors, aerodynamic contributions, and electromagnetic harmonics are the main sources of the machine noise [5,6]. Electromagnetic harmonic components in particular contribute a great number of high-frequency noises [7,8,9]. The electromagnetic noise is generated by the radial electromagnetic force harmonics in the air gap exciting the stator and the rotor. In order to reduce the electromagnetic noise in electrical machines, many researchers have proposed several methods, which can be classified into two categories:Passive noise reduction methods, which include1.Slot numbers combination [10,11];2.Machine geometry design [12,13,14];3.Machine PWM strategy [15,16,17,18];4.Passive anti-harmonic filter [19,20,21]; andActive noise reduction methods, which include1.Piezoelectric actuator [22];2.Injection of current harmonics [23,24];3.Injection of current harmonics in auxiliary windings [25];4.Active filter topology [26,27];

These methods can reduce the noise level in electrical machines, but they can also increase the difficulty of machine design and the complexity of control strategy, or they can add additional external power supplies. This paper presents a passive noise-reduction method based on a damper system consisting of a three-phase auxiliary winding in the stator slot and three external capacitors. This method has been applied to induction machines [28], but this paper extends this study to PMSMs. In fact, the stator damper winding has been widely investigated and applied in electrical machines, especially induction machines [29,30,31]. However, the objectives of these studies were to improve the power factor or reduce the unbalanced magnetic pull instead of reducing the machine noise. In this method, the auxiliary winding is wound in the stator slots and keeps the same winding distribution and star connection as the main winding. The three phases of the auxiliary winding are short-circuited via external capacitors. Owing to electromagnetic induction, the damping system can influence the noise signature of PMSMs by impacting the flux density harmonic components in the air gap. With suitable configurations, the damper system can achieve a damping effect on the electromagnetic noise. Without changing the internal design of machines and without the need for an external power supply, the damper reduced the noise by 7 dB in no-load experiments, achieving a significant damping effect on the PWM noise in particular. This research provides a different research direction and idea for the development of noise reduction technology in PMSMs.

The paper is structured as follows. In Section 2, the origins of electromagnetic noise in electrical machines are explained, focusing on the effect of flux density harmonics in the air gap. In Section 3, the paper describes the principle of the passive damper system and the equivalent circuits with the auxiliary winding are proposed for the theoretical analysis. The scientific approach and research process are also presented in this section. In Section 4, the impact of the damper system, particularly the capacitance value, on the magnetization currents is investigated with the help of the proposed equivalent circuits. Section 5 verifies and validates the theoretical analysis by means of the FEM (Finite Element Method). Section 6 presents the noise variation of a PWM-fed, 12-slot, 8-pole interior PMSM equipped with a damper system in no-load experiments, and the noise-reduction effect of the passive damper system is experimentally verified. Section 7 highlights the shortcomings of the damper system and the issues that need to be improved and optimized in the future. Finally, the research findings are summarized and the next steps of this research project are presented.

## 2. Origins of Noise and Vibrations

### 2.1. Flux Density Harmonics

The stator of AC machines creates a rotating field in the air gap that interacts with the magnetic field generated by permanent magnets, which leads to the torque production and the rotation of the rotor. Actually, the fields are not perfectly sinusoidal due to several reasons [5], such as the machine topology (e.g., slot/pole combination, plot shape, slot opening) [10,11,32], power supply [8,9,33], rotational speed, eccentricities [31,34], and magnetization [35]. The air gap flux density waveform $B_{\delta }$ can be decomposed as [2,36,37]
(1)$$\begin{array}{cc} B_{\delta }\left(t,\alpha \right) & =\left[\Theta_{s}\left(t,\alpha \right)+\Theta_{r}\left(t,\alpha \right)\right]\cdot \Lambda_{\delta }\left(\alpha \right)\\ \; & =\sum_{h}B_{\delta h}\left(t,\alpha \right) \end{array}$$
where $\Theta_{s}$/$\Theta_{r}$ are the stator/rotor magnetomotive forces versus position and time, and $\Lambda_{\delta }$ is the relative permeance of the air gap. The flux density harmonic amplitude ${\hat{B}}_{\delta h}$ is proportional to the magnetizing current $I_{\mu }^{h}$, where h should be for a given harmonic. A simplified expression is given [38] as
(2)$${\hat{B}}_{\delta h}=\frac{\sqrt{2}L_{\mu }I_{\mu }^{h}p}{N_{s}D_{is}L}$$
where $L_{\mu }$ is the magnetizing inductance, *p* is the pole pair number, $N_{s}$ is the number of turns in series per phase, $D_{is}$ is the inner stator diameter, and *L* is the length of the stator armature. The radial component and the tangential component of the hth flux density harmonic: $B_{\delta rh}$, $B_{\delta \Theta h}$ can be characterized by the pulsation $\omega_{h}$, angular position α, amplitude ${\hat{B}}_{\delta rh}$ or ${\hat{B}}_{\delta \Theta h}$, spatial harmonic number $r_{h}$, and initial phase $\phi_{h}$ in the 2p poles machine [39]:(3)$$\begin{array}{cc} \; & B_{\delta rh}\left(t,\alpha \right)={\hat{B}}_{\delta rh}\cos\left(\omega_{h}t-r_{h}\alpha -\phi_{h}\right)\\ \; & B_{\delta \Theta h}\left(t,\alpha \right)={\hat{B}}_{\delta \Theta h}\;\sin\left(\omega_{h}t-r_{h}\alpha -\phi_{h}\right) \end{array}.$$
$B_{\delta r1}$ and $B_{\delta \Theta 1}$ correspond to the fundamental components. There are an infinite number of harmonics in the air gap. Some of them induce electromotive forces in the stator winding. These electromotive forces generate other current harmonics and flux density harmonics.

### 2.2. Maxwell Forces and Noise

The interaction between the magnetic field of permanent magnets and the stator winding creates forces in the air gap. With the Maxwell stress tensor approach, the articles [40,41] explain how to deduce and simplify the expression of Maxwell forces in the air gap, considering several hypotheses. In fact, the noise and vibrations are mainly related to the radial magnetic pressure in the air gap $f_{h}=\frac{1}{2\mu_{0}}\left(B_{\delta rh}^{2}-B_{\delta \Theta h}^{2}\right)$ [36]. In the high-frequency range, the tangential flux density harmonics are much weaker than the radial flux density harmonics under no-load conditions. The radial magnetic pressure responsible for magnetic noise and vibrations can be determined [28] as
(4)$$\begin{array}{c} f\left(t,\alpha \right)=\sum_{h}f_{h}\left(t,\alpha \right)=\frac{1}{2\mu_{0}}[\sum_{h}B_{\delta rh}^{2}\left(t,\alpha \right)\\ +2\sum_{h}\sum_{h^{\prime }\neq h}B_{\delta rh}\left(t,\alpha \right)B_{\delta rh^{\prime }}\left(t,\alpha \right)] \end{array}$$
noting that $\mu_{0}$ is the vacuum magnetic permeability ($4\pi \times 10^{-7}\;H/\;m$). The force waves excite the stator and rotor mechanical structure, resulting in the noise and vibrations.

## 3. Investigation of Damper System

### 3.1. Three-Phase Damper System

The advantage of the damper system consists in installing an auxiliary winding, superimposed to the initial winding, in stator slots. All secondary stator coils of the same phase are in series with each other and with a capacitor. Star connection without grounding is used to link different phases. This auxiliary winding and the external capacitors form the damper system. There is no modification on the rotor nor the air gap. Figure 1 and Figure 2 show the position of the auxiliary winding in a concentrated winding PMSM with interior magnets.

In Figure 2, $v_{s}$ represents the supply voltage, $i_{s}$ or $i_{a}$ is the current in the stator or damper winding. The resistance, the leakage inductance of the auxiliary winding, as well as the capacitor, form a series RLC circuit in the equivalent single-phase diagram. Depending on the application, the chosen capacitor value allows adjusting or changing the resonant frequency of the auxiliary winding, thus affecting the filtering of current harmonic components.

This passive damper system allows the auxiliary winding to be installed without changing the internal design of machines. However, it also takes up space in the machine stator slots, increasing the overall weight and the space required for external capacitors.

### 3.2. Principle of Harmonic Reduction

The purpose of connecting capacitors is to allow flux density harmonic components ($B_{\delta h}$) to easily generate induced currents ($i_{a}$) in the auxiliary winding while maintaining a high impedance for low-frequency components, especially the fundamental component, whose induced currents must be weaker to avoid reducing the average torque. It must also ensure low joule losses to avoid reducing the machine efficiency, and in order to add only a low additional mass. The high frequency-induced currents in the auxiliary winding generate flux density components in phase opposition to those which create them, according to Lenz’s law. Therefore, specific flux density components at the origin of magnetic noise can be damped and reduced.

The interactions between flux density components and currents in the interior PMSMs are highlighted in Figure 3. As a rule, the fundamental component is directly involved in the generation of torque. Once the fundamental component has been reduced, it means that the average torque is reduced. This is why it is important to ensure that the fundamental component is not affected by the damper system. $B_{\delta h}$ can induce current harmonics in the auxiliary winding. But these current harmonics reproduce new weak flux density harmonics in the air gap. Overall, the flux density harmonics in the air gap are mitigated.

### 3.3. Equivalent Electrical Circuits

The synchronous machine with the damper winding can be represented in the d−q frame in Figure 4, where $\omega_{s}$ is the relative angular velocity between the stator α axis and the rotor direct axis in the electrical frame. In PMSMs, the magnetizing inductance is not always the same in the direct and quadrature axis, so the damping effect has to be separately studied on the two axes. Moreover, the electromotive force induced by magnets only exists in the *q*-axis circuit. For some machines with a special construction (e.g., surface-mounted permanent magnet machines), where the inductance is the same on the *d*- and *q*-axes, it is also possible to use only one equivalent diagram to explain the function of the damper, when only high-frequency harmonics are taken into account. In order to represent the damping effect, the current sign in the auxiliary winding is assumed to be opposite to that in the main winding (i.e., the phase difference is π). In fact, the direction of the damper current depends on harmonic frequencies and the reactance of the damper winding.

The Park transformation allows representing three-phase elements on the d−q frame, which is more intuitive. As with a general PMSM, the voltage equations can be expressed as follows [42,43]:(5)$$\begin{array}{cc} \; & v_{ds}=R_{s}i_{ds}+\frac{d\phi_{ds}}{dt}-\omega_{s}\phi_{qs}\\ \; & v_{qs}=R_{s}i_{qs}+\frac{d\phi_{qs}}{dt}+\omega_{s}\phi_{ds} \end{array}.$$

Among these equations, the magnetic flux linkage ($\phi_{s}$) is related to currents and magnets. Generally, the flux generated by magnets is considered as a constant on the *d* axis that does not vary with the stator current. As the damper is installed into the stator slots and uses the same winding distribution as the main winding, the corresponding equivalent circuits have to be derived to analyze the damper’s characteristics and performance. Considering the magnetizing inductance and leakage inductances of both windings, the equivalent electrical circuits with the damper branch are derived in Figure 5 from the voltage Equation (Equation 6):(6)$$\begin{array}{c} \left[\begin{array}{c} v_{ds}\\ v_{qs}\\ 0\\ 0 \end{array}\right]=\left[\begin{array}{cccc} R_{s} & -\omega L_{qs} & 0 & \omega \left(L_{q\mu }+l_{qsa}\right)\\ \omega L_{ds} & R_{s} & -\omega \left(L_{d\mu }+l_{dsa}\right) & 0\\ 0 & -\omega \left(L_{q\mu }+l_{qsa}\right) & R_{a}^{\prime }+\frac{1}{\omega C_{a}^{\prime }} & \omega L_{qa}^{\prime }\\ \omega \left(L_{d\mu }+l_{dsa}\right) & 0 & -\omega L_{da}^{\prime } & R_{a}^{\prime }+\frac{1}{\omega C_{a}^{\prime }} \end{array}\right]\left[\begin{array}{c} i_{ds}\\ i_{qs}\\ i_{da}^{\prime }\\ i_{qa}^{\prime } \end{array}\right]\\ +\left[\begin{array}{cccc} L_{ds} & 0 & -\left(L_{d\mu }+l_{dsa}\right) & 0\\ 0 & L_{qs} & 0 & -\left(L_{q\mu }+l_{qsa}\right)\\ L_{d\mu }+l_{dsa} & 0 & -L_{da}^{\prime } & 0\\ 0 & L_{q\mu }+l_{qsa} & 0 & -L_{qa}^{\prime } \end{array}\right]\begin{array}{c} dt\\ dt \end{array}\left[\begin{array}{c} i_{ds}\\ i_{qs}\\ i_{da}^{\prime }\\ i_{qa}^{\prime } \end{array}\right]+\left[\begin{array}{c} \omega \\ \omega \phi_{m}\\ 0\\ \omega \phi_{m} \end{array}\right] \end{array}$$
where *d* and *q* indices are corresponding components on the axis *d* or *q* respectively, $R_{s}$ the resistance of the stator winding, $l_{ss}$ or $l_{aa}$ the stator/damper leakage inductance, and $l_{sa}$ is the mutual inductance between stator and damper winding. $L_{ds}$ or $L_{qs}$ represents the total stator inductance on the *d* or *q* axis (i.e., $L_{s}=l_{ss}+L_{\mu }+l_{sa}$). $L_{da}$ or $L_{qa}$ means the total inductance of damper winding. For ease of understanding, it is assumed that the diagram of the equivalent circuits is based on the primary winding side, so the parameters of the auxiliary winding are expressed as relative values. When $R_{a}^{\prime }$ tends to infinity and $C_{a}^{\prime }$ is equal to zero, the classical electrical equivalent circuit of the PMSM can be obtained.

### 3.4. Scientific Approach

In order to scientifically evaluate the feasibility of damper system and the reliability and accuracy of the equivalent circuits, the following sections explain the research method and process in detail. The first step is to calculate the variation of magnetization currents by using the equivalent circuits, because it is positively related to the air gap flux harmonics as well as electromagnetic noise in PMSMs. The impact of damper winding on the magnetizing currents is investigated to predict the noise-reduction effect. Then, the FEM simulations are carried out to calculate the variation of air gap flux density harmonics with different capacitor values in the damper winding. This step can verify the magnetizing current curves estimated by the analytical method and can intuitively show the damping effect in machines. The next step is to experimentally verify the feasibility of the noise reduction by the passive damper system in a PMSM by measuring the noise level. In the experiments, a search coil in stator can also measure the variation of the air gap magnetic field to indicate the impact of the damper system. Finally, in this study, the findings from the different stages can be compared to evaluate the damper performance and the reliability of the analytical method. In Figure 6, the basic steps in the research process are shown.

## 4. Analytical Study of Damper System

According to the Equation (Equation 2), the noise and vibrations can be effectively reduced by minimizing the magnetizing current harmonics. Adjusting the impedance of the damper winding is a way to change current harmonics in two branches (magnetizing and damper branches). But the resonance of the RLC circuit is not negligible, because this phenomenon can amplify the magnetizing current harmonics resulting in an amplified noise. Determining the value of leakage inductances is important to choose an adapted capacitor to avoid the resonance.

### 4.1. Estimation of Inductance Values

FEA (Finite Element Analysis) [44] and the analytical method [42,45] are currently the most popular methods for estimating these inductances. But neither one is perfect. A major problem with these two methods is that the modeling of machines is always different from the real structure due to several assumptions. For the FEA, a 2D simulation provides faster results than a 3D simulation, but it requires one to determine the end leakage inductance analytically. These issues can result in large computational errors in the estimation of the leakage inductances. Moreover, the experimental method with the actual technology doesn’t allow us to easily measure the different leakage inductances separately. The best way to estimate the leakage inductances is by modeling the machine as accurately as possible under FEA.

The studied machine is a machine LSRPM (Leroy-Somer^®^ Permanent Magnet Synchronous Machine) with 12 slots and 8 poles. The damper winding has a copper diameter $d_{a}=0.4\;mm$, and that of the principal winding is $d_{s}=1\;mm$. The primary and auxiliary windings have the same number of turns ($N_{s}$ = $N_{a}$ = 58), thus $C_{a}^{\prime }=C_{a}$ in this case.

### 4.2. Analysis of Magnetizing Current Harmonics

After estimating the equivalent circuit parameters (Table 1) with the FEM in Flux (Altair Flux2D^®^) at T=50°C, avoiding the magnetic saturation of the stator armature, the expression of the magnetizing current can be written as an equation in terms of the voltage, inductance, resistance, and capacitance from Figure 5. The circuit analysis shows the magnetization currents over capacitor values under different frequencies $I_{\mu }^{h}\left(C_{a}\right)$. In order to compare the resonances, the same voltage amplitude and phase are used for different harmonics. In practice, the amplitude and phase of the voltages corresponding to the different harmonics are different, and the back-electromotive force cannot be neglected. The resonance as an intrinsic property of the equivalent circuit does not however change with external applied voltages or back-electromotive forces, so the simplified simulations are feasible. In this case, the curve $I_{q\mu }^{h}$ is almost the same as $I_{d\mu }^{h}$.

In Figure 7, the fundamental magnetizing current ($f_{s}=50\;\mathrm{Hz}$) is unaffected by the damper winding, as can be seen. Although the computational errors of the leakage inductances are unavoidable, the curves indicate the variation of the currents with the damping effect and resonance, which aids in selecting an adapted capacitor to mitigate the electromagnetic noise and vibration. With the increase of frequency, the damping effect becomes more obvious, and the resonance is also more pronounced. When the capacitor value is reasonably large, the curve is no longer influenced by the resonance, and the damping effect is steady and efficient. Several cases can be discussed as follows:$C_{a}=333\;nF$: The resonances of the magnetizing current harmonics around 6000Hz occur at $C_{a}=333\;nF$. There are slight increases in the magnetizing currents near 3000Hz.$C_{a}=4\;\mu F$: Most of the magnetizing current harmonics are differently mitigated. But the resonant peak of the current harmonic at 2000Hz appears.$C_{a}=10\;\mu F$: The damping effect for the majority of current harmonics is further intensified, and the curves gradually stabilize. The curve of the current harmonic at 1000Hz hits the resonant peak.

As it is shown in Figure 7, a greater capacitor value favors the damping effect for high-frequency harmonics. Although the impact on low-frequency harmonics is negligible, an excessively large capacitance value can also result in a slight decrease in the impedance of the damper winding for low frequency-components, which can cause the fundamental component to be affected. The cost and weight of capacitors should also be considered. Therefore, the choice of suitable capacitors depends on how the machines are used.

## 5. Finite Element Method Simulations

The purpose of finite element simulation is to verify the results of theoretical studies by introducing a PWM power supply. The transient simulations based on the FEM are carried out with Flux and Activate (Altair Activate^®^). The fundamental frequency $f_{s}$ is 50Hz, the corresponding speed is 750 tr/min. SPWM (Sinusoidal Pulse Width Modulation) three-phase voltage is generated by Activate, with the switching frequency $f_{swi}=3\;\mathrm{kHz}$. The machine, powered by a three-phase voltage inverter V=104.18V (phase-neutral voltage), is running under no load (i.e., the output torque is almost zero).

Figure 8 shows the spectra of the radial flux density harmonics $B_{\delta hr}$ on a fixed point in the air gap with or without the damper winding. Their values are listed in Table 2. The radial flux density harmonic components, especially high-frequency components from SPWM signals at $f_{swi}\pm 2f_{s}$ or $2f_{swi}\pm f_{s}$ (e.g., 2900/3100Hz, 5950/6050Hz) that contribute greatly to the electromagnetic noise, are strongly impacted by the addition of the auxiliary winding. Different capacitors also affect the damping effect on different harmonic components:$C_{a}=333\;nF$: There are a few flux density harmonics that are slightly amplified. But the fundamental flux density component is stable. As with the previous analytical curves, $C_{a}=333\;nF$ is closer to the resonant peak of the flux density harmonics around 6000Hz, so the auxiliary winding amplifies these harmonics more obviously than it does around 3000Hz. Compared to the fundamental component, these variations are relatively small.$C_{a}=4\;\mu F$: The damper makes a significant harmonic reduction while having little effect on the fundamental component. The harmonic reduction effect is almost identical to the theoretical prediction.$C_{a}=10\;\mu F$: For the harmonics around 3000Hz and 6000Hz, the damper with 10μF capacitors achieves an even better damping effect than the damper which uses 4μF capacitors. This can be explained with the magnetizing current curves (Figure 7). The reason for this is that this capacitance value is further distant from the area where electrical resonances of these harmonics are likely to occur.

The results of FEM simulations corroborate the findings of a great deal of the previous work in theoretical analysis in Section 4.2. The variations of the flux density harmonics are comparable with the magnetizing current curves, considering the impact of the damper winding. As the electrical resonances can occur, they lead to an enhancement of the current harmonics passing the magnetizing inductance, thus increasing the flux density harmonics in the air gap. On the contrary, the damper with suitable capacitors can realize the PWM harmonic reduction in the air gap flux density.

## 6. Experimental Results

Experiments are used to verify the findings of simulations and theoretical studies. The same voltage source as simulations is applied to the studied machine prototype and the same PWM strategy ($f_{swi}=3000\;\mathrm{Hz}$) is used. The machine runs under FOC (Field Oriented Control) [46,47,48] at 50Hz under no-load operation. Three capacitors (333nF, 4μF, 10μF) are respectively used in experiments.

### 6.1. Global Sound Pressure Level Measurement

Figure 9 illustrates the global sound pressure level in four cases with different capacitors. In the case with $C_{a}=333\;nF$, the sound pressure level is always slightly higher than in initial system. With $C_{a}=4\;\mu F$ and $C_{a}=10\;\mu F$, the sound pressure is lowered on average by more than 5dB and 7dB, respectively. This finding broadly supports the theoretical work in Section 4.2 and also accords with the previous simulation results, which show that the damping effect is realizable but the noise amplification can also occur with an inappropriate capacitor value in the studied PMSM.

### 6.2. Acoustic Noise Spectra

Figure 10 compares the sound pressure spectra with different capacitors in the one-third octave frequency band, in which the variation of noises can be more visually compared. The four experimental cases can be separately discussed:

Case 1: In the initial system without damper, the noises around $f_{swi}$ and $2f_{swi}$ are more important than other noise components, which indicates the electromagnetic harmonics contribute significantly to the machine noise under this operating condition.Case 2: The damper winding is connected to the capacitors with $C_{a}=333\;nF$. The noises around 6000Hz are increased by 10dB. For these noises, a positive correlation is found between the electrical resonances of magnetizing currents in Section 4.2 and the noise-amplification phenomena in the experiments. Around 3000Hz, the impact of the damper is inconspicuous. In the experiments, there is a slight decrease in noise in this range, whereas theoretical studies show a slight rise in these noise components. However, considering the error of the noise measurement and theoretical computation, the difference is tolerable and reasonable. For the noises with frequencies above 10000Hz, this damper still has a good noise-reduction effect.Case 3: For $C_{a}=4\;\mu F$, the significant noise reduction can be observed in the noise spectra and in the global sound pressure level in Figure 9. In particular, the PWM noises are effectively mitigated. It can be seen that the noises around 2000Hz are amplified, compared to other cases, which is also consistent with the previous curves in Figure 7, because in this case, the electrical resonances of current magnetizing occur. Fortunately, this variation contributes slightly to the global noise level.Case 4: With $C_{a}=10\;\mu F$, the damper works better, because there are fewer noise-amplification phenomena. In the low-frequency band, the small increase of noises does not affect the global noise level, but the high-frequency noises are obviously reduced. Similar results can be seen in the magnetizing current curves (Figure 7).

Comparison of the findings with those of theoretical studies and FEM simulations confirms that the noise reduction by the passive damper system is feasible and the analysis by the equivalent circuits is effective.

### 6.3. Electromotive Force Spectra

By using a search coil, an independent single-phase winding following the path of one turn of a stator phase, the variation of the magnetic flux inside the machine is measured as an induced voltage signal. In Figure 11, it can be clearly observed that the amplitude of the fundamental component is almost stable in different cases. On the contrary, as with the simulation results, the harmonics at high frequencies are strongly influenced by the auxiliary winding.

With $C_{a}=333\;nF$, the amplifications of flux density harmonics occur around 6000Hz. But it is more visible than the noise spectra in Figure 10 that the flux density harmonics around 3000Hz are slightly amplified by the damper, which is consistent with the trend of the magnetizing current curves and the results of the simulations. The case with $C_{a}=10\;\mu F$ has a better damping effect than that with $C_{a}=4\;\mu F$. The electromotive force spectra corroborate the finding in the simulations (Figure 8). And they further support the theoretical studies with the equivalent circuits.

## 7. Discussion

As it is shown in this study, although the damper significantly reduces the noise of the PMSM, some shortcomings deserve to be discussed:Additional mass and volume: The damper wound in stator slots increases slightly the system weight. The superimposed auxiliary winding takes up slot space, thus lowering the fill factor of the main winding. The added mass and volume are difficult to avoid, but it is possible to minimize their impact by optimizing the damper winding configuration, according to operating conditions.Amplification of current harmonics: In the theoretical study and experiments, the amplification of current harmonics occurs in the stator winding. The problem can be explained by the equivalent electrical circuits in Figure 5 and Figure 7. The damper reduces the current harmonics in the magnetizing inductance and thus the noise of the machine. Most of the current harmonics pass the damper side. This is in line with the objectives of this study. But for the electromagnetic harmonics, the whole system has a lower total impedance, because of the addition of the damper branch. Thus stator current harmonics ($i_{s}=i_{a}+i_{\mu }$) in the principal winding are inevitably increased. On the one hand, the harmonic currents may enter the electrical grid and on the other hand, the energy efficiency ratio of the machine is reduced. Considerably more work will need to be done to determine the variation of stator current harmonics under different operating conditions.Increase of power consumption: In the experiments, the machine with the damper consumes more electrical energy. The extra power may depend on the damper configuration and PWM parameters. It is probably related to the amplification of the stator current harmonics. The additional power consumption can be investigated later to see if it varies with the increase of mechanical load.Limitations of damper system: At present, this paper only verifies the PWM noise reduction by damper system, but further research is needed to study its impact on other electromagnetic noises. For example, the back-electromotive force harmonics are closer to the fundamental frequency, compared to the high-frequency harmonics from PWM, so it is difficult for the damper system to effectively reduce the noise from these harmonics without affecting the fundamental components.

## 8. Conclusions

This study has shown that the damper winding can successfully and effectively reduce the electromagnetic noise in a no-load PWM-fed PMSM. In particular, in the experiments the global sound pressure level is reduced by more than 7dB. To efficiently reduce harmonic components, the appropriate capacitors also need to be used; otherwise the damper winding could have an amplification effect on specific harmonics. Using capacitors with the largest possible capacitance value is a way to avoid resonances. The fundamental component is not affected by the damper system with correct capacitors in general, but it is necessary to ensure that the auxiliary winding still maintains a high impedance for low-frequency components. Furthermore, the experimental results are in accordance with the expectations of the theoretical calculations and simulation results, which corroborate the accuracy and reliability of the new proposed electrical equivalent circuits with the damper branch. In order to improve the application of the passive damper system, it is necessary to optimize the damper configuration considering not only the noise reduction effect, but also the operating conditions of machines, the manufacturing costs, and the shortcomings of the damper. Load experiments will be conducted in future studies to verify whether the dampers have a negative effect on the output torque. More in-depth research can then be carried out to improve the application of the method in different scenarios.

## Figures and Tables

**Figure 1 sensors-22-02738-f001:**
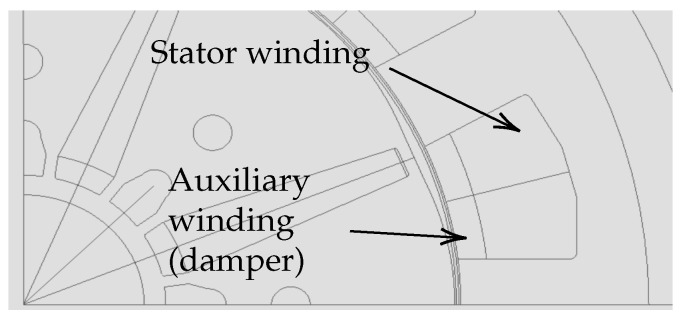
Modelling of a PMSM with the auxiliary winding.

**Figure 2 sensors-22-02738-f002:**
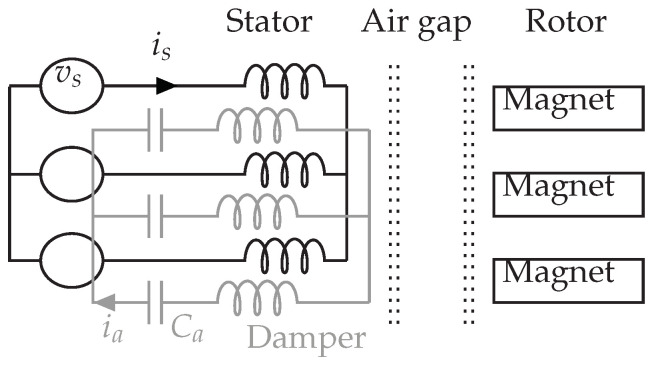
Representation of the damper winding.

**Figure 3 sensors-22-02738-f003:**
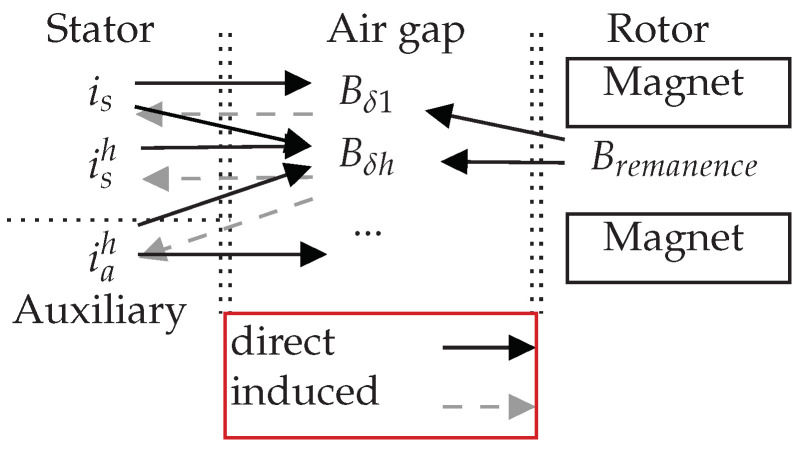
Interactions between the stator and rotor.

**Figure 4 sensors-22-02738-f004:**
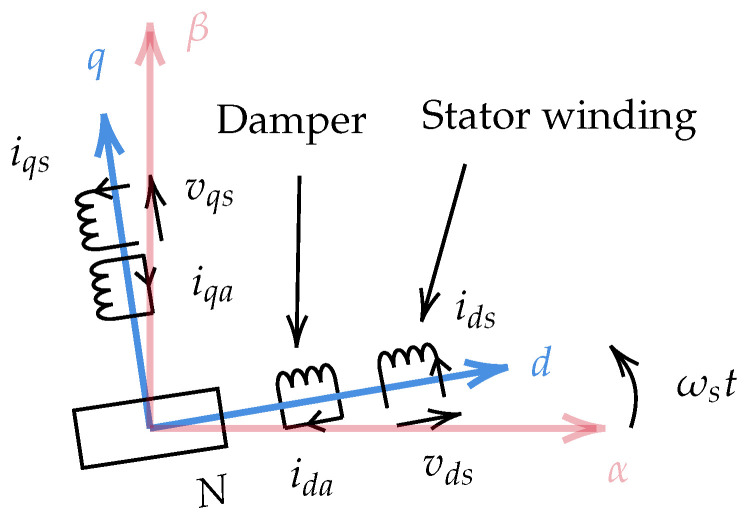
Representation of a PMSM in the d−q frame.

**Figure 5 sensors-22-02738-f005:**
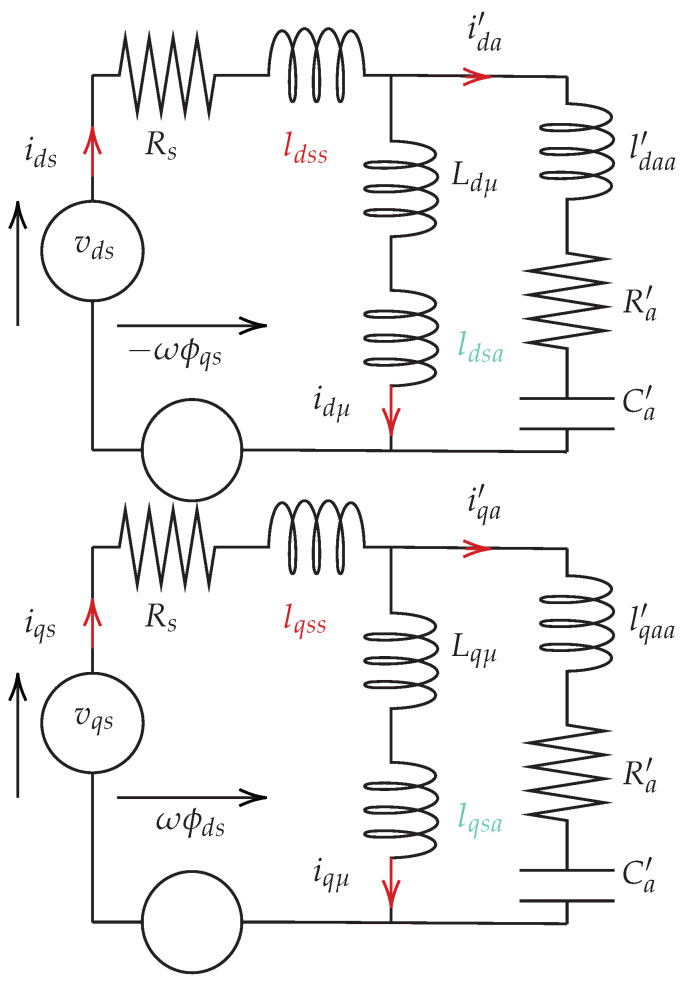
Equivalent electrical circuits on the primary winding side. In the *d*-axis (**top**). In the *q*-axis (**bottom**).

**Figure 6 sensors-22-02738-f006:**
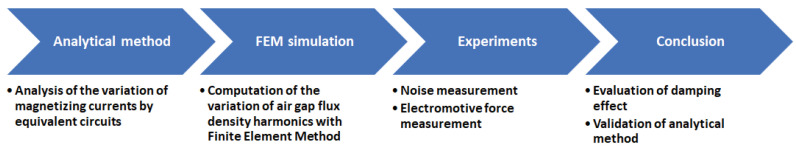
Flowchart of the research process.

**Figure 7 sensors-22-02738-f007:**
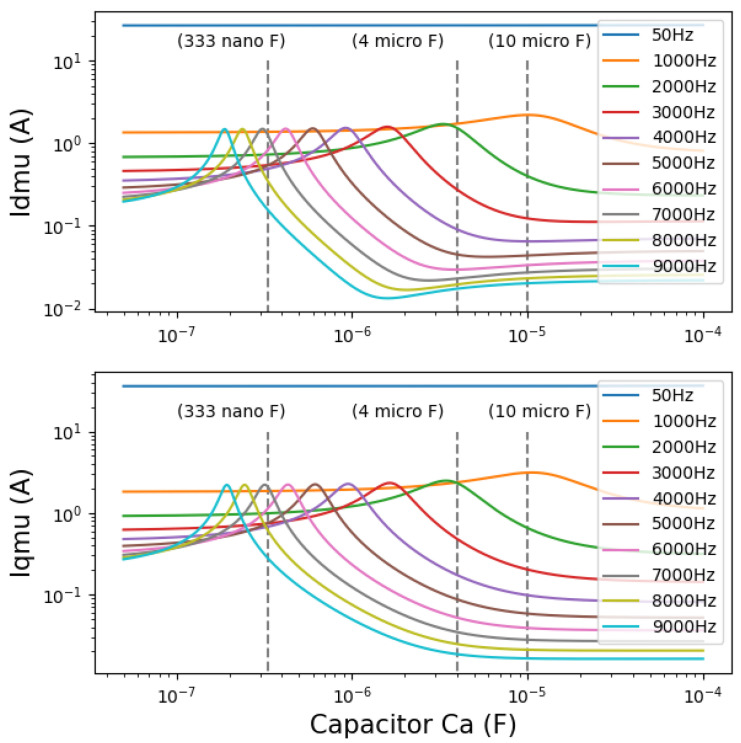
Magnetizing currents versus capacitor values under different frequencies estimated by analytical method. Magnetizing currents in the d-axis ($I_{d\mu }^{h}\left(C_{a}\right)$, **top**). Magnetizing currents in the q-axis ($I_{q\mu }^{h}\left(C_{a}\right)$, **bottom**).

**Figure 8 sensors-22-02738-f008:**
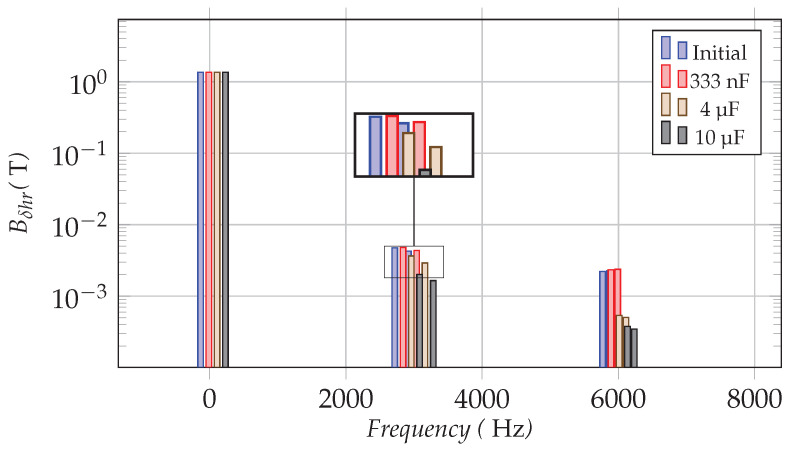
Radial flux density spectra ($B_{\delta hr}$); initial system and with different capacitors $C_{a}$ in the damper.

**Figure 9 sensors-22-02738-f009:**
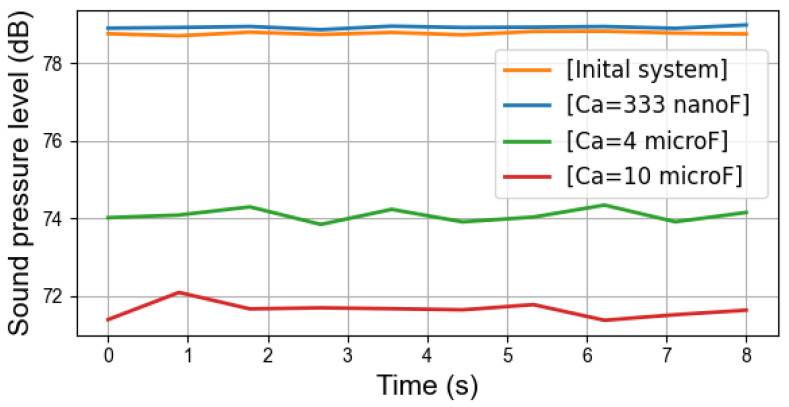
Global sound pressure level in dB with $f_{swi}=3\;\mathrm{kHz}$.

**Figure 10 sensors-22-02738-f010:**
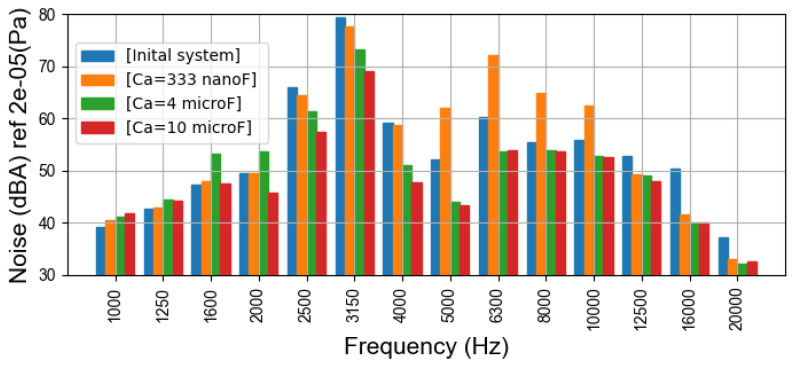
One-third octave spectra of the sound pressure in dB with $f_{swi}=3\;\mathrm{kHz}$.

**Figure 11 sensors-22-02738-f011:**
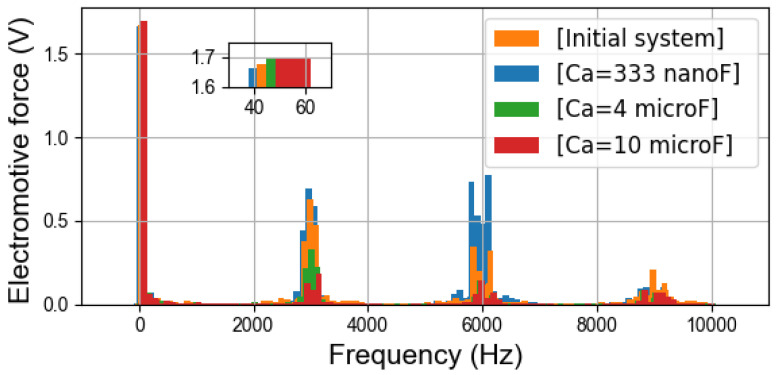
Electromotive force (one turn) spectra with $f_{swi}=3\;\mathrm{kHz}$.

**Table 1 sensors-22-02738-t001:** Equivalent electrical circuit parameters, T=50°C, no saturation.

Estimated with FEA		
$l_{dss}$	1.52	mH
$l_{qss}$	1.71	mH
$L_{d\mu }$+$l_{dsa}$	27.89	mH
$L_{q\mu }$+$l_{qsa}$	20.16	mH
$l_{daa}$	0.21	mH
$l_{qaa}$	0.021	mH
Estimated analytically		
$R_{s}$	1.16	Ω
$R_{a}$	7.24	Ω

**Table 2 sensors-22-02738-t002:** Radial flux density harmonics in the air gap $B_{\delta hr}$(T).

f(Hz)	Initial	333nF	4μF	10μF
50	1.3537	1.3537	1.3549	1.3567
2900	0.004735	0.004809	0.003641	0.002013
3100	0.004260	0.004345	0.002909	0.001647
5950	0.002213	0.002338	0.0005357	0.0003779
6050	0.002256	0.002376	0.0005028	0.0003462

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
