# Peer review of "Damper Winding for Noise and Vibration Reduction of a Permanent Magnet Synchronous Machine"

_sensors, 2022, doi:10.3390/s22072738_

Round 1

Reviewer 1 Report

This paper addresses an interesting issue, i.e., the reduction of the emission of noise and vibration by a Permanent Magnet Synchronous Machine (PMSM).

Although this paper is very technical and it is quite difficult to follow each part of it, I think that the results are interesting and worth to be published in Sensors. The noise emission reduction is about 7 dB in the most efficient configuration of the damping system and this achievement is relevant from the point of view of machinery noise emission.

The study is well described and scientifically rigorous, the results of an analytical method are compared to those of a numerical one and, finally, the results of an experiment are presented.

The damper for noise reduction are shown not to introduce effect on the fundamental harmonics which mainly affects the average torque of the PMSM.

Minor comments: I included some minor comments in .pdf paper attached

Author Response

Dear reviewer, 

Thanks a lot for your comments and suggestions. Firstly, I am very honored by your recognition of our research. As in your comments, we have modified and improved these issues. Considering that MDPI Sensors is a very rigorous and high-quality journal, we have also rewritten the introduction, the scientific approach, and the results. We hope they will help you to better understand our study. We look forward to hearing from you! 

Best regards

Reviewer 2 Report

The specific comments are as follows.

1) The Introduction needs to be rewritten. It should contain background, motivation, detailed Literature review, novelty and contribution. Currently, the paper significantly lacks such structure.

2) What is new in this paper? The concept of using damper winding has been there for a long.

3) Show the design steps using a flowchart and explain it.

4) The results are not well articulated. Please divide it according to various scenarios and explain.

Author Response

Dear reviewer, 

Thank you very much for your review. I am honored to have your comments on my paper. They are important and helpful to improve our paper and research. According to your suggestions, we have seriously rewritten several sections,  especially the introduction, the flowchart of the research process, and the results. Subsections are added to improve the clarity.

For your questions, we can reply here. Of course, we added some corresponding explanations and references in the introduction of the paper:

About the novelty of the damper winding: Yes you are right, the stator damper winding has been widely used in electrical machines, especially in induction machines. But their objectives are to improve the power factors or to reduce the unbalanced magnetic pull. Our research focuses on the noise reduction by the damper winding in permanent magnet synchronous machines. A similar method has been investigated in induction machines in our laboratory. At present, this study is extended to PMSMs. This can help us to better understand the effect of damper windings on noise in different machines. It is important to optimize the damper configurations to fit different scenarios.

I hope my answer meets your expectations. 

We look forward to hearing from you. 

Thanks again,

Best regards!
